# Towards an Understanding of the Pre-War Landscape Transformations in the Face of Contemporary Urban Challenges on the Example of Gajowice in Wrocław

Aleksandra Gierko 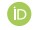

Department of Public Architecture, Basics of Design and Environmental Development, Faculty of Architecture, Wrocław University of Science and Technology, 50-317 Wrocław, Poland; aleksandra.gierko@pwr.edu.pl

**Abstract:** This paper discusses the results of desk and field studies conducted in the Gajowice estate in Wrocław. The aim of the paper is to identify the original assumptions of the development of areas around multifamily buildings and to examine the process of their transformation to the present day. The research hypothesis states that the used solutions would now be defined as green infrastructure or nature-based solutions. This was confirmed with the help of comparative cartographic studies. Research on the original land development of the interwar period allows for identifying the principles based not only on compositional aspects, but also the recognition of natural values in the variety of green forms used in a given area and the important role of trees with large target sizes, in addition to the principle of shaping the green system that permeates the urban tissue, creating ecological corridors and positively influencing the local climate. Thus, the historical development is in line with the contemporary postulates of climate resilient cities.

**Keywords:** greenery; land development; green areas; green infrastructure

## 1. Introduction

The rapid growth of the European urban population since the nineteenth century has become a canvas for new urban concepts in which greenery has been playing a significant role. The regulatory plan of Berlin, developed under the supervision of James Hobrecht, became a guideline for other plans within the German Empire's cities, including Wrocław. Established in 1862, the so-called Hobrecht Plan assumed to solve the problems of overpopulation and the hygiene of the inhabitants' lives. A basic city block module was accompanied by a large courtyard that helped to ventilate the flats and allow sunlight to enter buildings. The planners' assumption was also to introduce green areas into the cityscape [1].

The first attempts to establish and implement zoning plans in Wrocław at the beginning of the twentieth century were slowed down by the First World War, which was followed by the Spanish flu epidemic, and an economic crisis only intensified the problems of the overpopulated city. The response to this instance in the form of well-thought-out urban solutions in the 1920s and 30s was the result of strategic planning by the city authorities in combination with the activities of construction societies and architects who understood the need to implement modern solutions.

Almost a hundred years after the housing estates' construction, the urban tissue was transformed, also as a result of the damage caused during the Second World War. Nevertheless, historical urban thought can still be read, especially with the use of comparative studies. The uniqueness of the architecture built in the interwar period was noticed by post-war specialists, and its building complexes were documented and recognized as a cultural heritage, mainly in terms of buildings, but not through the prism of landscape or greenery. Meanwhile, the study of the period's city planning rules allows one to see the holistic concept of shaping urban fabric with greenery. In this aspect, greenery shall be

treated as a historical green infrastructure, influencing not only the local environment and microclimate, but also the wellbeing of city dwellers.

At present, increasingly frequent studies show that the availability of green areas, including allotment gardens, near places of residence affects the physical and mental wellbeing of residents [2,3], which can also be associated with the social properties of public spaces [3,4]. The principles of urban landscape planning introduced intuitively by urban designers over a hundred years ago are now reflected in ideas such as One Health. According to this approach, the health of humans, animals, and the entire ecosystem are interconnected [5]. Thus, the city is to be a biodiverse ecosystem, in which greenery plays not only an aesthetic role, but also provides ecological connections for animals and their habitats. Such a multi-layered attitude towards the role of the green system in the urban fabric is currently being promoted by the European Green Deal initiatives. The European Biodiversity Strategy was drafted under this plan's umbrella. It emphasises the role of areas created as a result of natural processes and the necessity of their conservation, but also draws one's attention to green urban spaces as places that provide a number of ecosystem services, both for humans and other organisms [6]. In the case of greenery, it is particularly important to introduce it into the urban space at different scales and so that it creates a network of connections that permeates the cityscape [7] (pp. 65–69). Land development adjacent to housing estates built in accordance with the state of knowledge at the time, in response to welfare and health problems, could be a reference for contemporary solutions combining design, ecology, and social and economic accessibility.

This paper discusses the results of desk and field studies conducted in the Gajowice estate in Wrocław and is a part of the broader research of the spatial development of pre-war housing estates. The aim of the paper is to identify the original assumptions of the development of areas adjacent to multifamily buildings, bearing in mind the wider urban planning context of the interwar period, and to examine the process of their transformation to the present day. Because of the numerous changes that land development has undergone, the original assumptions are now difficult to read using direct observation. The research hypothesis states that due to the adopted design principles, these estates were planned with great care for the buildings' surroundings; therefore, the used solutions would now be defined as green infrastructure or nature-based solutions. Thus, the historical development is in line with contemporary postulates of climate-resilient cities. Solutions examined with the help of comparative cartographic studies could become an introduction to a catalogue of local solutions, mitigating climate change, supporting biological diversity and human health that is also based on cultural heritage.

## 2. Materials and Methods—Scope and State of Research

As the study is a part of a broader research encompassing several housing estates in Wrocław, a spatial and thematic delimitation of the study was made for the purposes of this paper. The temporal scope of the issue is the same as for the entire study and concerns housing estates from the moment of their construction, i.e., from the interwar period to the present day. However, studies on urban tissue provenance date to the beginning of the nineteenth century. The spatial extent, shown in the figure below (Figure 1), was determined by the thematic scope. The subject of interest of the study was the development of areas adjacent to multifamily buildings, which jointly possessed the following features:

- Apart from general spatial dispositions, it was possible to recreate the details of land development on the basis of archival materials;
- The development showed the features of a broader urban concept;
- Post-war transformations did not completely erase the pre-war plan.

The spatial scope (Figure 1) encompasses a part of the Gajowice housing estate in Wrocław, whose border runs along present-day Krucza, Kwaśna, Grochowa, Jemiołowa, Krucza, Gajowicka streets, Generała Józefa Hallera Avenue, and a part of Wrocław's railway bypass. Contemporary Polish place names have been consistently used in this paper.

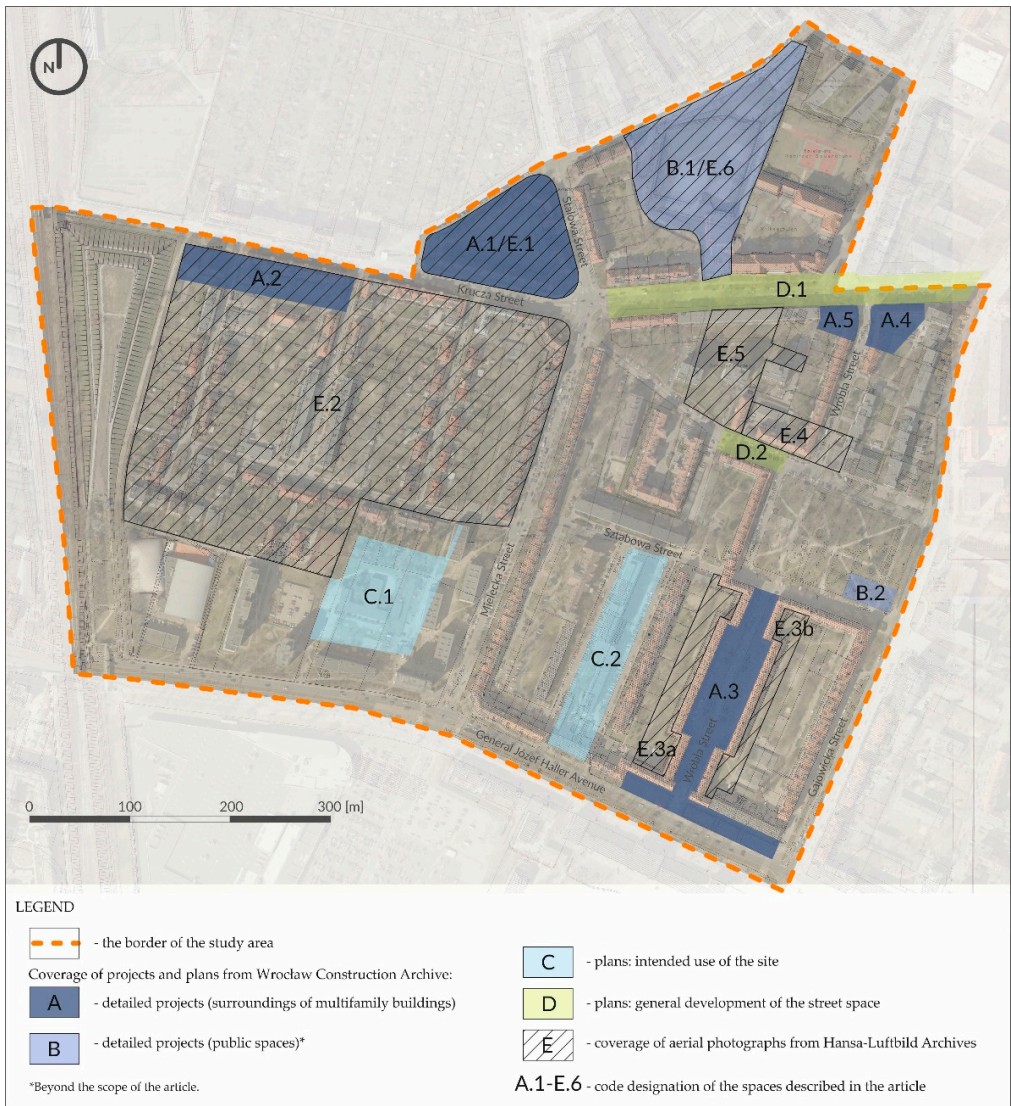

**Figure 1.** Spatial scope of the study with the coverage of the iconographic and cartographic sources. Original work based on historical cadastral maps [8–10] and contemporary aerial photography from 2018.

Among the used research methods, two groups shall be distinguished: chamber studies and field studies. The essence of chamber research was the collection and appropriate development of cartographic, iconographic, and literature materials, which could then be used to outline the study background and comparative cartographic analyses that were then supplemented by field studies. Comparative analyses, commonly used in landscape studies [11,12], were aimed at assessing the dynamics of landscape transformation and assessing the state of preservation of pre-war land development. The comparison of pre-war cartographic materials and architectural drawings with the current state was made in the GIS. Materials in the form of raster files were georectified: a geographical reference was added to them on the basis of control points. Map overlaying for tracking landscape changes is widely used in studies at different scales and for different purposes [13–15]. When analysing changes, attention was also paid to factors that determined the manner of land rearrangement related to period trends, thus referring to the method of landscape biography [16]. An important aspect of the field study was a detailed familiarisation with the examined objects and the confrontation of the actual state of preservation with the performed chamber studies. The field research was carried out using the method

of direct observation combined with photographic documentation in August 2019 and February 2021.

The study background was drawn on the basis of a review of written sources from the pre-war period: a book publication [17], an article in a sector literature [18], and a local law act with its graphic appendix [19,20]. The first post-war publications from the 1950s concern general issues of Wrocław's urban development [21]. Embedding local tendencies in a broader context was possible thanks to a review of topical publications [22–29]. The current neighbourhood profile was outlined based on the city scale planning document [30], a consultation report [31], and an overview of participatory budget projects [32–34]. However, all these sources have their limitations, as mentioned in the subsection.

The contemporary research on architecture and its spatial arrangement during the interwar period in Wrocław was studied by, inter alia, Wanda Kononowicz [35]. Nevertheless, the spatial transformation of the estate's greenery has not been studied before. Single entries regarding the development of the areas covered by the article can be found in the Green Lexicon of Wrocław. Present access to cartographic and iconographic sources, including modern digital data, allows for in-depth comparative research and tracing the changes that green areas have been subjected to. The sources were obtained as a result of a query of archives in several institutions, which are summarised in Table 1. The list shows where a given document is stored, their typology, and dating, as well as an overview of their accuracy in terms of the land development study.

**Table 1.** A summary of iconographic, cartographic, and modern digital data sources used for the purposes of the study in the article.

| Location | Source Type [1] | Dating | Description [2] |
|---|---|---|---|
| Wroclaw University Library | large-scale, manually drafted plans [20,36] | 1865 and 1926 | general urban layout; cadastral division, road system, layout of buildings, watercourses, and overall land cover record in the plan from 1865 [36]; division into building classes in the plan from 1926 [20] |
| Wrocław Construction Archive of Museum of Architecture | pre-war manually drafted plans and designs collected by City Magistrate and Construction Police [8–10,37–44] | 1911–1943 | depending on source; plans [8–10] with cadastral division, road system, layout of buildings, and overall land arrangement records such as water, trees, and terrain; plans [37–39] with general site purpose (urban disposition, future public areas); designs [40–44] with details of land development (playgrounds, benches, pavement materials, single trees, low planting types) |
| Municipal Water and Sewerage Company S.A. in Wrocław Archive | pre-war manually drafted sewage system plans [45,46] | 1912 and 1932 | design [45] and plan [46] of the sewage system with a site plan for buildings, road network, and land development |
| Herder Institute for Historical Research on East Central Europe | aerial black and white perspective photographs from the Hansa-Luftbild Archives [47–56] | 1929–1934 | land development details (elements of playgrounds, individual trees, low planting types), land use (school gardens, utility yards, sports grounds, etc.) |
| Military Historical Bureau in Warsaw | aerial black and white orthogonal photographs [57–60] | 1947 | land development details (greenery system) and the scale of post-war damage |

**Table 1.** *Cont.*

| Location | Source Type [1] | Dating | Description [2] |
|---|---|---|---|
| Main Office of Geodesy and Cartography in Warsaw | aerial black and white [61–64] and colour [65] orthogonal photographs | 1974, 1985, and 1995 | depending on the extent and season; photographs from 1974 and 1995: urban development on the scale of the greenery system (it is possible to identify individual trees); photographs from 1985: details of land development (playgrounds, etc.), also under tree canopies |
| Wrocław Spatial Information System | property map and orthoimagery [66] | 2015–2021 | cadastral division [66] and details of land development in 2015 and 2018 [66] |
| other online resources | pre-war iconography [67] | no exact date | details of land development: type of greenery and appearance of elements such as benches |

[1] References in square brackets. [2] Including the degree of accuracy.

## 3. Results

### 3.1. Historical Background of the Study

After the First World War, Wrocław was one of the most populous cities of the Weimar Republic. Its population density reached 114 people per hectare of the total urban area, while the country's 46 major cities had an average number of 41.3 people per hectare [17]. This was mainly due to the small amount of land occupied by built-up area when compared to cities of an even smaller population, such as Frankfurt am Main. The post-war issue of housing shortages in the cities of Weimar Republic caused a critique of the free market and its role in shaping poor living conditions. It resulted in establishing affordable housing construction programmes on the national level. Low-profit enterprises were to solve the problems of not only accommodation, but also unemployment [23]. It was possible also due to rationalization of architectonic forms and building process. Technology was to improve living conditions, as perceived by architects [26,28].

The above-mentioned free market critique also involved issues of cityscape. In 1919, landscape architect Leberech Migge published 'Green Manifesto' in which he developed his idea of planning productive landscapes within the city. Migge was influenced by the Garden City Movement, but upscaled this idea to the whole country that was to become a garden. He perceived the connection of a city and land as a way to healthy society: not only in terms of physical health, but also economic self-sufficiency and freedom [24]. These ideas were transferred to the narratives of architects, such as Bruno Taut, who emphasized the essence of common access to land in his theoretical treatise and implementations [27]. In later years, Leberech Migge incorporated architectural thought into his considerations, writing about the important role of light in the animal and plant world, especially in temperate climate. In the concept, he proposed locating residential spaces on the southern side of the building [29].

The Garden City Movement itself, however, had influenced Berlin's planning policies before the First World War. The 'Greater Berlin' plan by Hermann Jansen was the first attempt to limit the spontaneous development of the city. The green belt around the built-up areas created an urban climate system and has given them a spatial order [22]. Jansen's plan remained a theoretical model until administrative reform and expansion of the city in the 1920s. In 1929, his postulate gained a real dimension through the provisions of the first urban development plan, prepared under the direction of Martin Wagner. Wagner saw in green spaces particular social and public health values. Under the influence of those ideas, the city authorities, through programmes implemented in the 1920s, increased the availability of recreational areas for a large number of residents and made green sites occupied about 20% of the urban area [25].

The municipal authorities of Wrocław decided to solve the overpopulation problem by implementing a wide-ranging programme of building affordable flats, combining public activities with those of purposively founded construction companies that dealt with not

only building up individual plots of land near the pre-war tenement buildings, but also the construction of entire estates. The areas attached to the city in 1868 were largely developed. The beginning of the 1920s spurred the construction of vast housing estates, such as Sępolno or Popowice, with an extensive functional and spatial programme. Those ventures were the responsibility of the Siedlungsgesellschaft Breslau joint-stock company, whose activity stood out among those of other building societies of interwar-period Wrocław. Low-rise housing estates, mostly two-and three-storey, uniformly shaped, were built. Care was taken to provide adequate flat insolation, ventilation, and size. Two-and three-room flats constituted the majority, i.e., almost 90% of all units [17]. The activities of building societies were characterised by a planned organisation and economic approach, visible in the standardisation of apartments and building elements. The societies were also obligated to carry out certain projects in public areas, including the construction of underground utility grids, planting street trees, and wall climbing plants [35]. Moreover, famous Wrocław architects were invited to design, which had a significant impact on the appearance of the housing estates. They were built in the spirit of Modernism, using new technologies, following a holistic approach to shaping the urban fabric, including green areas. Framing with greenery was perceived by the architects of the time as an activity that influenced the harmonisation of the building with the surroundings [18]. A holistic urban planning approach, modern architectural forms, and technological solutions, such as central heating and a shared steam laundry—first introduced in 1926 in the block limited by what are now Krucza, Kwaśna, and Stalowa streets—made it so that the estates of the interwar period did not fall behind those built in other contemporaneous cities of the Weimar Republic, such as Schillerpark in Berlin designed by Bruno Taut.

The appropriate proportions of green sites in relation to built-up areas were strictly regulated by law. In the 1920s, a new zoning plan was developed for Wrocław, based on pre-war ordinances from 1912. According to the new building code, a part of the plot had to remain undeveloped—it could serve as an internal courtyard or garden, but it could not be used for utility functions, such as fuel depots or animal pens [19]. At present, this could be referenced to areas designated for parking lots or waste disposal sites. In those regulations, an emphasis was placed on the health-related aspects of living: unit insolation and ventilation. Buildings higher than two storeys had to have yards with an area of no less than 80 m$^2$ [19]. The entire city was divided into building classes, with a greater share of green areas in newly built housing estates compared to the older city centre. In the area of today's Gajowice, the standard was 6/10 of the undeveloped plot area [19,20].

The crisis of 1929 slowed down the pace and momentum of construction projects. Political changes and the necessity to deal with another war continued this trend. Despite this, post-war Wrocław, thanks to the actions of pre-war urban planners, could boast a high percentage of two- and three-room apartments compared to other cities within the borders of the new state [21].

*3.2. General Pre-War Assumptions and Implementation of the Spatial Development of Gajowice Greenery*

At the time of including today's Gajowice site within the boundaries of Wrocław, in the 1860s, the areas on which the current housing estate is located were arable lands belonging to the village of Gabitz, located on their western side. The map from 1865, an excerpt of which is presented in Figure 2, shows the road and water system as well as the division of arable land [36]. An important element of agricultural areas was the so-called Sour Source (Sauerbrunn in German)—a water reservoir created at the site of a water exudate, probably surrounded by greenery, which at the time of urban regulation became a canvas for arranging a central recreational space in the district. Part of the road system was respected when designing the urban plan [37]. It became, among other things, the basis for the delineation of the block limited by the present-day Krucza, Kwaśna, and Stalowa streets.

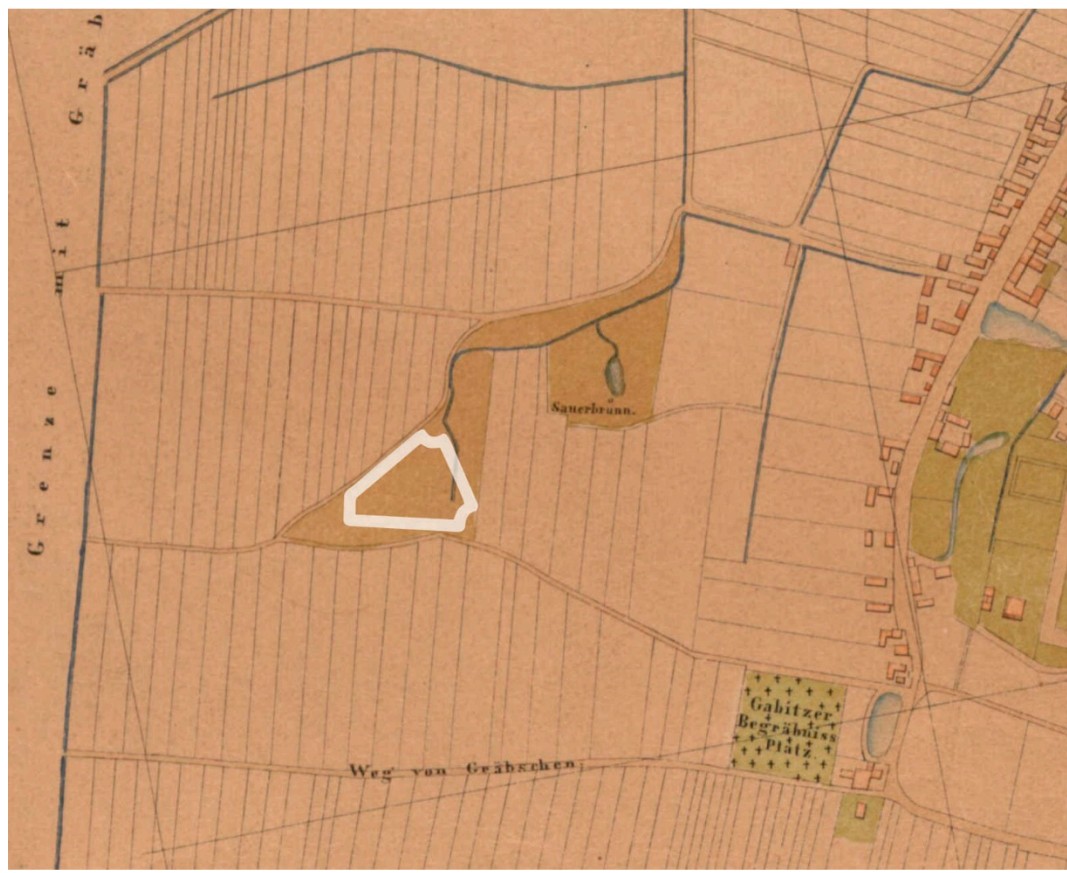

**Figure 2.** Indicative location of the development line of the quarter limited by the present streets of Krucza, Kwaśna, and Stalowa (white line) on the excerpt of the plan from 1865 [36]—original work. Field roads became the basis for marking out the street system.

The urban plans of the district were created at the turn of twentieth century. The site was developed naturally from the city centre, from the north-east, gradually towards the open urban fringe. The area of interest for this article developed mainly in the 1920s and 1930s, although regulatory plans were drawn up prior to the First World War. Then, the formal character of the intersection of Krucza, Stalowa, Mielecka, and Bernarda Pretficza streets was planned. The arrangement of buildings around the intersection was to create a square space. The urban layout was to be closed on three sides, while the corner at the intersection of Krucza and Bernarda Pretficza streets was to remain undeveloped and intended for a formal green space. Newly planted trees were to be an important element of street space, especially the double lane running in the middle of Mielecka Street [45]. After the First World War, the plans were verified: Mielecka Street was marked out at a different angle and in a smaller width, in the 1930s, buildings were introduced in the area that previously had been planned for a green square. The urban tissue that had been destroyed during the next war was later rebuilt in the same layout, as shown in Figure 3.

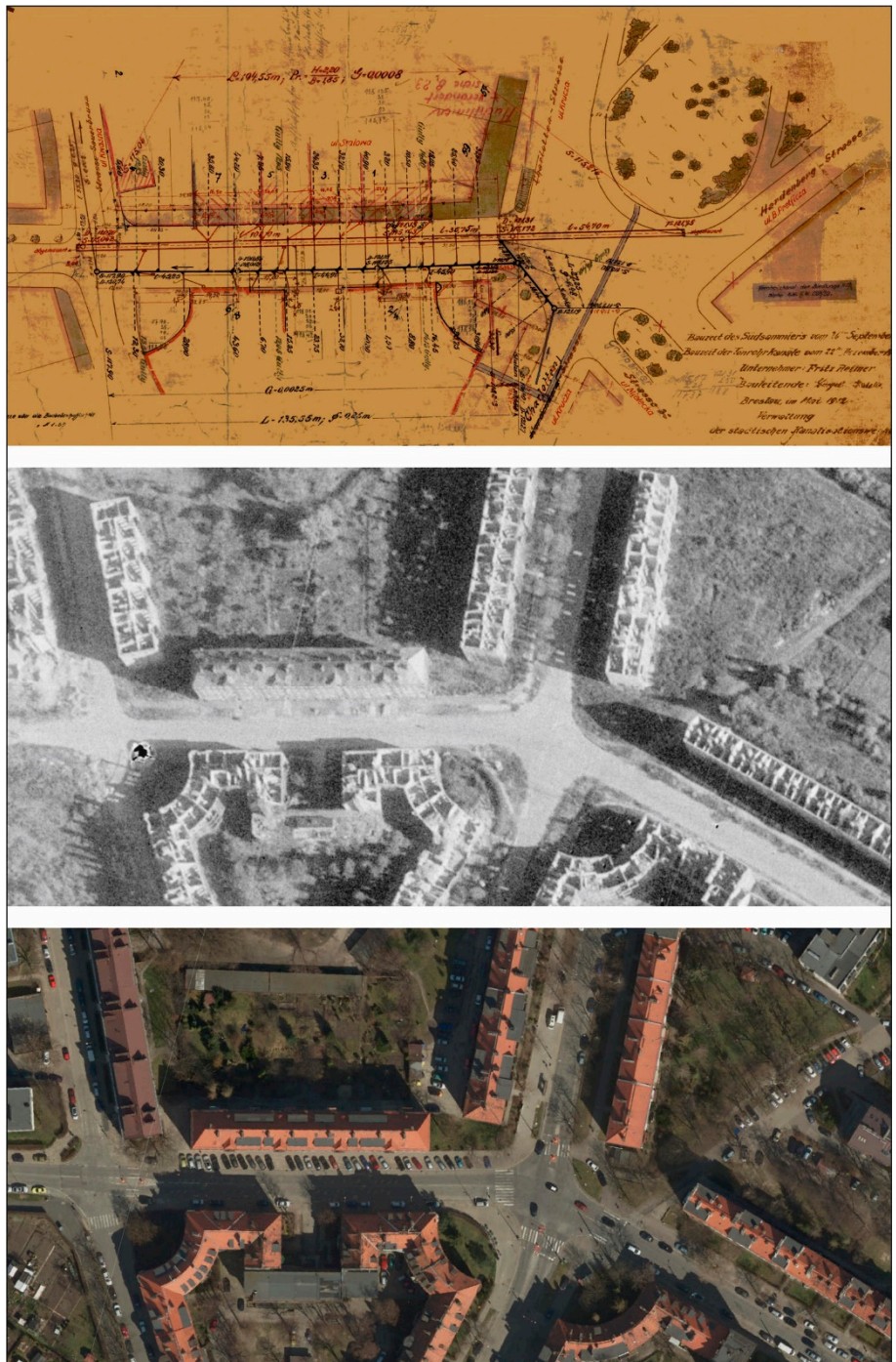

**Figure 3.** Development of the intersection of Stalowa, Krucza, Mielecka, and Bernarda Pretficza streets: a juxtaposition of a part of the project from 1912 [45] with aerial photographs from 1947 [60] and 2018 [66].

At the turn of the 1920s and 1930s, a tram line was built along Krucza Street (D.1 in Figure 1) and ended at the above-mentioned intersection [46,52]. Krucza Street was the main artery of the estate leading from east to west: from Gajowicka Street towards Grabiszynek. Its width was measured at 30 m between building lines. The cross-section at the height of Wróbla Street, going from the north towards the south, was as follows:

- sidewalk, 6.23 m wide;
- greenery with a row of trees, 0.50 m wide;
- bicycle road, 1.50 m wide;

- roadway, 8.00 m wide;
- bicycle road, 1.50 m wide;
- hedge, 0.50 m wide;
- tram track, 5.60 m wide;
- hedge, 0.80 m wide;
- sidewalk, 5.37 m wide [46].

It was a well-furnished street, with wide pavements and bicycle paths on both sides of the road. The tram track was separated from the carriageway using hedges. A row of trees, probably lime trees, was continued on the northern side of Krucza Street up to the railway embankment [9,60]. The southern part of the street from Mielecka Street to the embankment was probably treated as a reserve for the extension of the tram line. Street trees were an important element of land development. Noteworthy is the strip of trees located at Bernarda Pretficza Street (D.2 in Figure 1), at the height of Wróbla Street, which was established between two carriageways [9,57], currently with a mixed species composition and the last relic planting of catalpa trees.

Perhaps the most important public greenery site in the estate was the area covering a significant part of the block encircled by Krucza, Stalowa, Kwaśna, Grochowa, and Jemiołowa streets. The park at Sour Source (B.1/E.6 in Figure 1) was established at the turn of the century on the basis of a rebuilt pond that collected spring water. Surviving documentation from many archival sources would allow for a careful study of the area, almost no trace of which has survived to the present day, and could be subject to separate research, similarly to the playground at Gajowicka Street (B.2 in Figure 1). Temporary recreational areas were also depicted in pre-war aerial photographs. The photographs from 1932 and 1934 show a complex of tennis courts adjacent to the present Bernarda Pretficza Street (E.4 in Figures 1 and 4) [52,55,56], which in the 1940s was transformed into a built-up area adjacent to Wróbla Street.

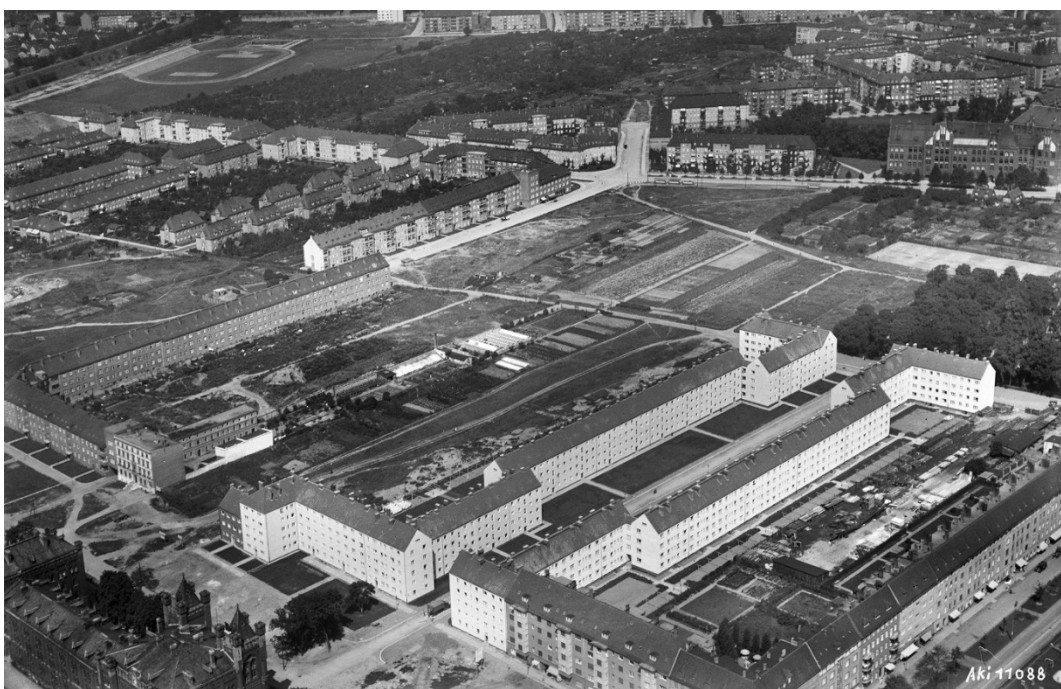

**Figure 4.** The aerial photograph taken above Generała Józefa Hallera Avenue towards the north-west in 1932. Buildings along Wróbla Street in the foreground. In the upper right corner, a school on Krucza Street and school gardens on the other side of the street. Next to it, tennis courts (bright area on the right) [52].

The aerial photographs also show large areas designated for cultivated gardens adjacent to buildings, which provided part of the household food supply. Sites that were

not yet built-up at the time were intended for temporary allotment gardens, as in the case of much of the land lying in the quarter of Stalowowolska, Połaniecka, Mielecka Streets, and Generała Józefa Hallera Avenue (C.1 in Figure 1) [38]. Before the war, the school at Jemiołowa Street had an extensive school garden stretched between Krucza and Bernarda Pretficza Streets (E.5 in Figures 1, 4 and 5), a small part of which was built up in the 1940s with a house perpendicular to the Bernarda Pretficza Street [57]. The garden was probably a place for students' education and the basis of their school meals. Tall trees were planted along the border of the lot. The alleys of trees marked the internal divisions of the garden, and the main axis of the composition was an avenue that ran across the green area from north to south [55]. The garden was liquidated after the war, and new residential buildings were erected in its place at the end of the 1960s, thus blurring the entire development, including trees. The only relic is probably a magnificent oak growing at the back of houses on Wróbla Street. Another example of the completely different needs of the post-war years are two plots limited by Generała Józefa Hallera Avenue, Buska, Sztabowa, and Stopnicka Streets (C.2 in Figures 1 and 5) that were private and mainly used as a horticultural farm. In the 1940s, they were designated for public green areas with a playground and a promenade [39], in line with the systemic approach to greenery and city policy pursued since the turn of the twentieth century, to buy private plots of land and transform them into parks and squares. The area was still used for the production of plants until the 1970s [61]; however, in the 1980s, it was built over by erecting an eleven-storey multifamily building. The current manner of development of the pre-war green areas is shown in Figure 5.

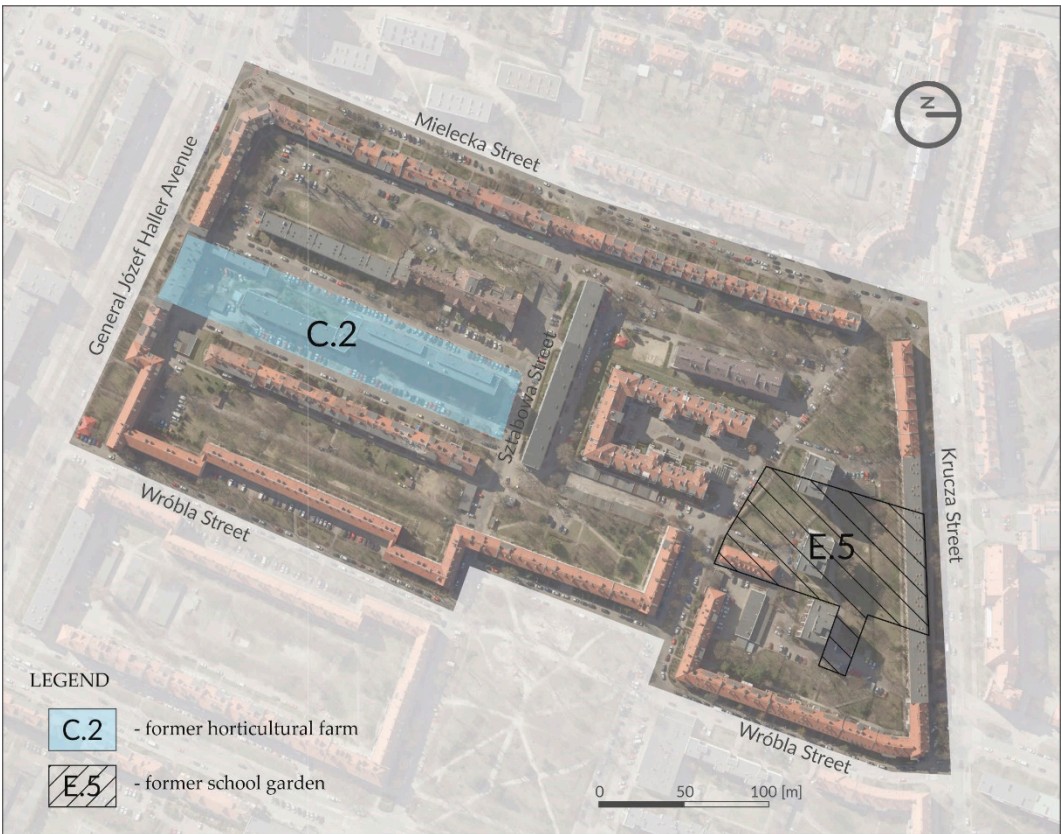

**Figure 5.** Pre-war green areas of the land designated for a public green space (C.2) and school garden (E.5) within the pre-war cadastral divisions [9,10] plotted on aerial photography from 2018 [66].

### 3.3. The Development of Areas Belonging to Multifamily Buildings

The main focus of the study is areas adjacent to multifamily buildings. These sites were intended for the everyday needs of residents. Nevertheless, in a broader planning perspective, the interwar designers noticed their significant impact on human health. In the preserved materials from Wrocław Construction Archive of the Museum of Architecture [40–44], compared to other cartographic and iconographic sources and the current state, a wider urban context taken into account in those projects can be seen. The modern systemic approach to greenery proves its societal and environmental potential.

The first case under study is the interior courtyard of a residential building in the triangle of Krucza, Kwaśna, and Stalowa streets (A.1/E.1 in Figure 1). As mentioned before, the plot was marked out according to the course of field roads (Figure 2). As seen on a map from 1865, the plot had been located between two roads, and a watercourse had run along the present Stalowa Street, bending to the east and joining Sour Spring [36]. The regulatory plan from 1911 still showed the canvas of field roads that marked the course of the present segments Kwaśna and Krucza Streets [37]. In fact, the course of the former dirt road leading to the Grabiszynek village had been visible even in 1929, before the actual construction of the road occurred and was immortalised on an aerial photograph, documenting the erection of the housing estate on the southern side of Krucza Street [48].

The estate on both sides of this part of Krucza Street (A.1/E.1 and E.2 in Figure 1) was given the name 'Am Sauerbrunn', which referred to its location and can be translated from German as 'At Sour Spring'. The contemporary name of Kwaśna Street (Sour in English) is a reference to this name. The buildings constructed in the triangle of Krucza, Kwaśna, and Stalowa streets were the first to be heated via central heating in interwar Wrocław. The entire complex, with a large, coherent garden in the interior surrounded by buildings predominantly with three-room apartments, was designed by a well-known and respected architect: Hermann Wahlich. The implementation was published in the trade magazine at the time [18], and it was also well documented in aerial photos of the interwar period [47–49,52–55]. A drawing showing the land development plan has been preserved in the Wrocław Construction Archive [40].

Aerial photographs allow the assessment of the actual pre-war use of the space between buildings and the verification the accuracy of archival drawings. The photographs were found to be consistent with the drawing deposited in the Wrocław Construction Archive and at the same time to indicate the use of presumably preliminary concepts as an illustration in a press article. A photograph of a mock-up of a building block published in a trade magazine shows the general site plan of the surroundings [18]. From the side of Krucza Street, front gardens bordered by low hedges were designed, while in the courtyard, an interior road running along the facades of buildings, a strip of low greenery separated by hedges and a line of trees surrounding the central square were planned. The layout of the courtyard's interior repeats the triangular outline of the facade. The published drawing shows perhaps the first concept of a site development plan, which differs from the mock-up and the actual state. The designed outlines of green patches are softer and more organic compared to the resulting simple geometric development drawing. The entrances to the centre of the yard are accentuated with pairs of trees, and in the centre, there is a double row of trees.

The completed project from 1927, shown in Figure 6, assumed hedging the front gardens and accentuating the entrances to the building with plantings on both sides of the entrance, located on the pavement side [40]. Based on individual trees preserved in these places, it can be assumed that they were hawthorns of a pink blooming variety. The aerial photograph from 1932 [53] shows spherically shaped small trees, which may confirm this hypothesis. Trees of the same size are visible at the entrances to the building on both sides of the gateway from the side of Stalowa Street. The photograph also shows a row of trees—probably lime trees—planted along the northern part of Krucza Street, which was a section of a wider concept that was described earlier. However, the archival drawing does not include the green strip between the bicycle path and the pavement. Based on

comparison of the drawing with the iconographic sources and the current measurements, it can be assumed that the cross-section from the building wall to the road edge could be as follows:

- front garden, 5.0 m wide;
- sidewalk, 2.5 m wide;
- greenery with a row of trees, 1.5 m wide;
- bicycle road, 2.0 m wide.

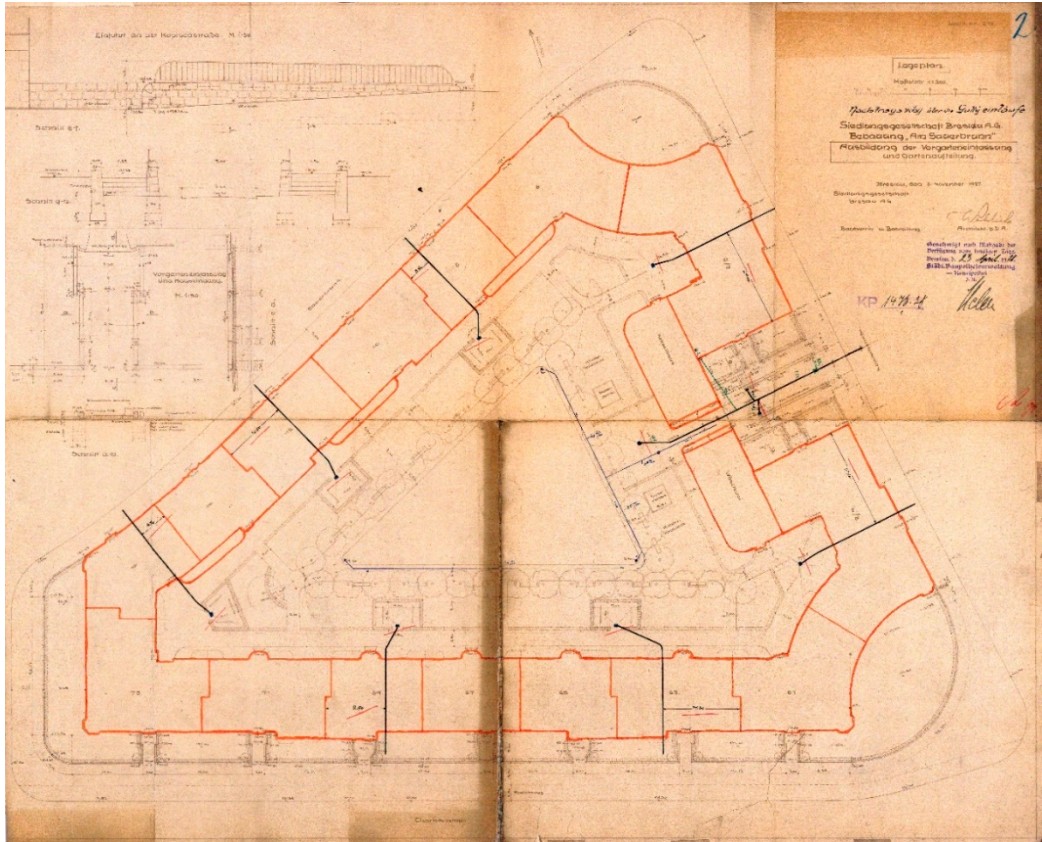

**Figure 6.** Implemented courtyard design from 1927 by Hermann Wahlich. The buildings outline marked with red colour. Drawing is north-east oriented [24].

The facades of the buildings shown in the photograph described above are mostly covered with climbing plants, both from the street and the courtyard sides (Figure 7). A comparison of this image with almost the same shot from three years before (1929) shows a significant growth of greenery, including climbers, hedges, and trees [48]. The corner in the form of an inverted arch, located at the intersection of Krucza, Stalowa, and Mielecka streets, overgrown with creepers and additionally planted shrubs of different heights in a well-thought-out composition emphasising the facade form and the formal character of the street's intersection, looks particularly impressive.

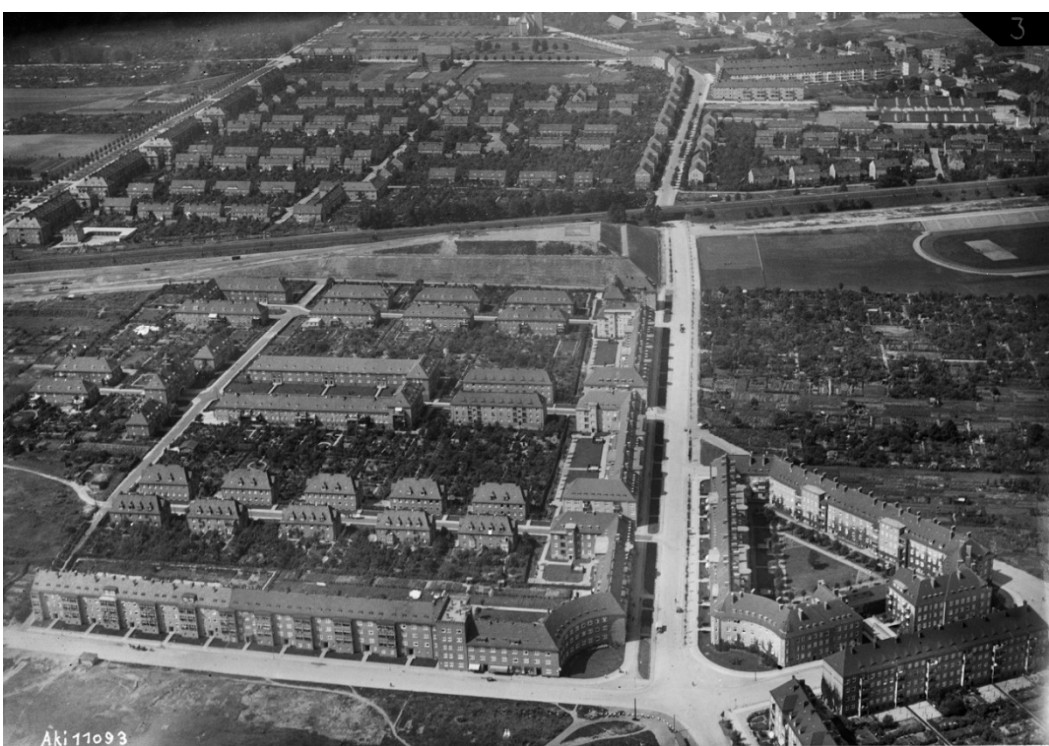

**Figure 7.** The aerial view at 'Am Sauerbrunn' estate along Kurcza Street towards the west in 1932. On the right is the triangle courtyard designed by Hermann Wahlich. On the left side of the street are wide lawns—a probable reserve for tram line extension [52].

The implemented design of the courtyard assumed 2.5 m wide pedestrian circulation spaces along the internal northern and southern elevations [40]. On the west side, there was a gateway leading to a yard with a square outline, closed on the sides with tall, probably hornbeam, hedges. There, symmetrically located on both sides of the square, were separate playgrounds with sandboxes and benches. Along the remaining two elevations, there was a strip of low greenery, 8 m wide, with hornbeam hedges encompassing utility boxes with carpet hangers. In several places, access was provided to the central triangular part that was underlined with a line of lime trees, along which there were bench niches. The inner square with a triangular outline, covered with grass, was separated from the surroundings by paths, and in its centre, a solitaire was planted, emphasising the composition. The interior design was simple, and it resulted from rectangular geometric forms. Greenery was treated as an urban material that complements the architecture.

The preserved aerial photographs show an additional, compared to the design, planting of bushes on the corners of the central square, which is also confirmed by an undated photo taken from the level of a pedestrian [67]. It was probably made in the 1930s and shows perennial beds emphasising the bypass along the buildings and simple forms of benches with concrete legs and wooden seats and back supports. On the basis of the photography, it could also be supposed that the internal pathways were constructed with gravel mixture, so that they had a partially permeable surface.

Post-war transformations in the development of the area under study can be traced via comparisons of aerial photographs taken in 1947 [60], 1974 [61], and 1985 [63] and present-day orthoimagery [66]. The photograph taken after the war shows the damage done to the buildings, while the composition of the tall greenery, both street and courtyard trees, was preserved. Bearing in mind not only the transformation of the site, but the broader context of the area under study, the Gajowice estate, it would be crucial to trace the changes by verifying materials from the 1950s and 1960s. However, there is substantial probability such that such materials do not exist. Nevertheless, it was at this time that

much of the urban fabric of the estate began to be systematically supplemented and rebuilt. At that time, the Sour Source was presumably also covered.

A photograph from 1974 [61] recorded the state of the interior in the triangle of Krucza, Kwaśna, and Stalowa streets after rebuilding. The road system in the courtyard was reorganised—a 2.5–3 m wide circular road was introduced, which was paved with concrete blocks. The road was moved away from the facade of the building and separated from the wall with a strip of greenery and sidewalks. The permeable surface under tree canopies was narrowed to a width of about 3.5 m. This circulation system has remained to this day. In the 1970s, the tall greenery was still complete and compact. In the 1980s, only two trees remained from the street row, and the roadside lane began to be intensively used as a parking space for cars. The photograph from 1985 [63] also shows the functional layout that has been preserved to this day: the waste collection point on the axis of the entrance from the side of Stalowa Street and the playground in place of the central lawn.

Until today, only 23 limes remain out of the 41 trees planted before the war. The courtyard interior is a parking place for residents, which also has a significant impact on the high greenery condition (Figure 8). The greenery of the middle floor has grown at different times, and there are no intentionally composed shrubs or perennials. The custom of keeping creepers on the walls of buildings has not been continued. The composition of the space emphasises the utility functions of the interior, which had been supposed to serve the recreation of residents according to the original concept. Nevertheless, a large number of permeable surfaces and large trees constitute the natural and cultural value of this place.

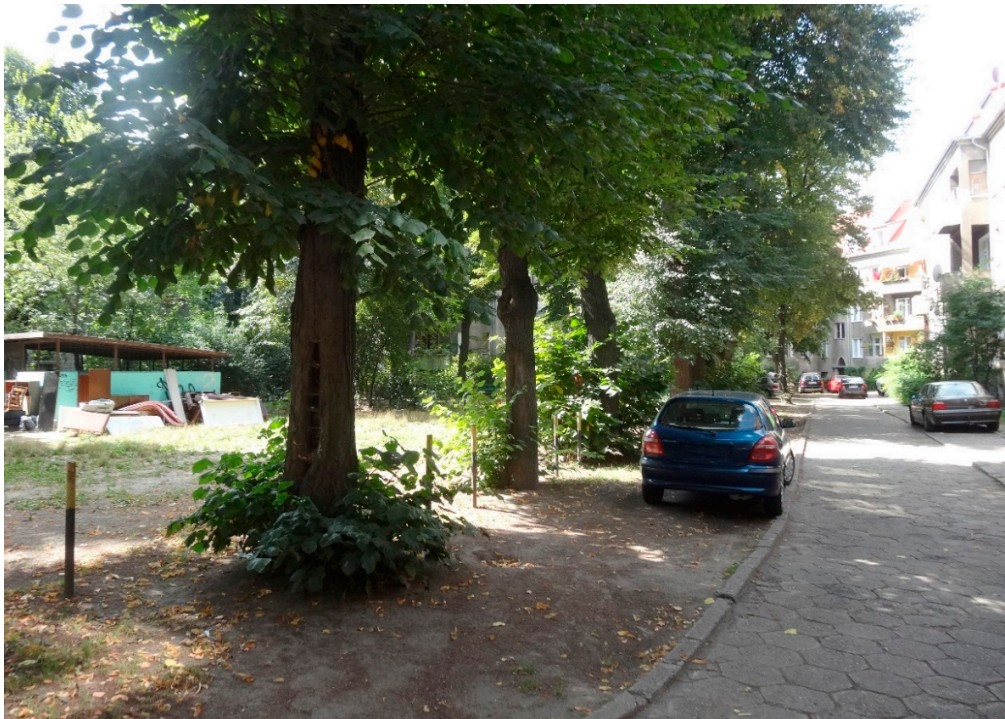

**Figure 8.** The current state of land development: the obliteration of the historical layout and the use of the courtyard for utility purposes. Parking under trees adversely affects their condition.

The second case analysed in this study is a part of the estate established in the same period at the opposite side of Krucza Street. The entire urban layout marked E.2 in Figure 1 could be studied on the basis of aerial photographs [47–55]. The houses located in today's Kolbuszowska, Stalowowolska, Tarnobrzeska, and Połaniecka streets were built mostly in the mid-1920s. Due to the low development density—only a few families lived in each house—the adjacent areas were arranged as home gardens. The gardens had ornamental

functions, their layout was often geometric, based on one or two axes, and they served a utility function. In accordance with the assumptions prevailing at the time, which were also used in the no-longer-existing housing estate in Popowice in Wrocław, most of the gardens had tool sheds shared by residents. The public front gardens had a simple geometric layout, and the trees were used only in a double lane planted in a small green area at the end of Kolbuszowska Street. Shared semi-public courtyards were laid out in a sequence of interiors adjacent to the buildings constructed along Krucza Street (Figure 9).

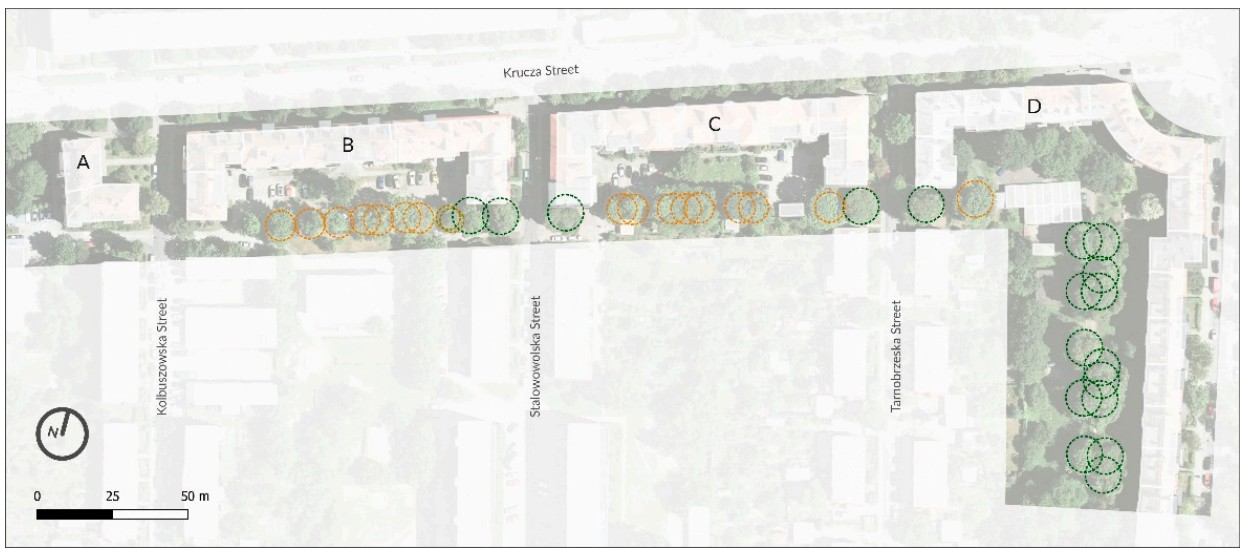

**Figure 9.** Courtyards with relics of historical greenery in the buildings' interior along Krucza Street. Chestnut trees marked in green, hornbeams in yellow, and maple trees in orange. Plotted on aerial photography from 2015 [66].

The buildings were erected in 1928–1929 as a project by Siedlungsgesellschaft Breslau. The buildings marked with the letters A and B in the figure were designed by Heinrich Rump, and buildings with the letters C–D by Hermann Wahlich. Concerning site plan drawings, only Heinrich Rump's drawing has been preserved in the Wrocław Construction Archive [41]. However, on the basis of comparative analyses of iconographic and cartographic resources with site studies, it is possible to reconstruct the idea of landscape development. Building A has an L-shaped plan. Such an arrangement of buildings determined that the remaining part of the plot was divided into two sections, each with a different character [41]. The front part was a formal entrance to the building, with a pavement running along its walls and a square-shaped greenery site in the middle. The green area was surrounded by a hedge, and the corners were planted with shrubs accentuating the composition. The part behind the building served recreational and utility functions. The courtyard was fenced, with two gates leading to it and an exit directly from the building. The western part, separated from the public promenade by a hedge, had a space with a sandbox and a U-shaped bench around it, and a separate spot with a carpet hanger (Figure 10).

The space behind building B, also designed by Heinrich Rump, was set on the north–south axis, perpendicular to the building and dividing the backyard into two, almost symmetrical parts. The front gardens were fenced with hedges, and the entrances were accentuated with bushes or hawthorns in stem form. The yard had the shape of an extended rectangle, about 20 m deep and about 80 m long. Directly next to the building, a 2.5 m wide green strip was marked, separated by access routes to the entrances. There was a path around the yard, probably permeable, 2.5 m wide. The middle part consisted of two large lawns, separated by a 2.0 m wide passage located on the north–south axis. From the side of the building, the passage was arranged in the form of a small square, 6.0 m wide, separated from the space by a hedge. The corners on both sides of the lawns near buildings were also separated by hedges and featured carpet hangers. At the end of the

courtyard, in the southern part, there were two sandboxes and benches mirrored on both sides of axis. Six trees were planted on each side of the southern fence. On the basis of field studies and measurements, it can be concluded that maple trees were planted here, and the distance between them was 4.5 m. At the corners of the building, on the south side, bench niches were arranged on the projection of a part of an arch, planted with formed hedges. On the west side, the niche closed the yard with 3/4 of the arch, and there were two benches there, while on the eastern side, the niche was a half-arch with one bench. A passage was to be made to Stalowowolska Street, towards the east. In reality, however, as recorded in the aerial photographs [51,53,55], a westward exit was made instead. The same photographs also show that the southern facades were planted with climbing plants. On the basis of field studies of the preserved trees, it can be added that the hedges at the bench niches were planted with hornbeams. A pair of chestnut trees was planted at the side of the unrealised entry. Upon inspection in the field, it can be stated with certainty that this has become a spatial rule of this part of estate: the streets were flanked by plantings of chestnut trees that were closing the rows of maples planted inside the courtyards, as shown in Figures 9 and 11.

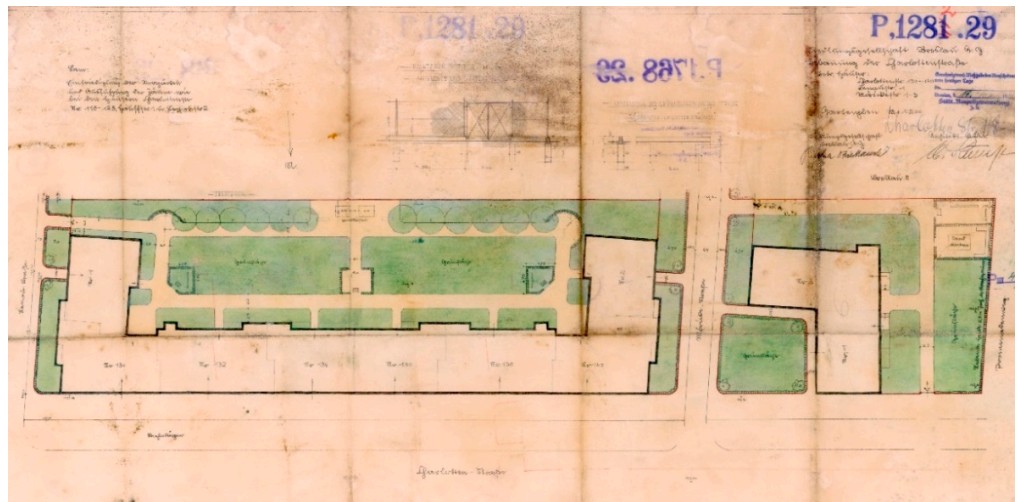

**Figure 10.** Implemented land development plan for buildings along Krucza Street from 1929 by Heinrich Rump. Drawing is south oriented [41].

The interior of the courtyard at building C was realised as a mirror image of the courtyard at building B, with an exit towards Tarnobrzeska Street, which is recorded in the aerial photographs [53–55]. Maple trees were planted at shorter distances from each other (4 m), so there were seven or eight trees on each side of the axis. In addition, the southern facades were planted with climbing plants. The series of interiors was closed by a fenced yard at building D. The character of land development refers to the general principle applied throughout all yards, which means that despite the divisions, both by street and fences, the sequence of backyards could be interpreted as a coherent assumption, both from a functional and environmental perspective. The yard closing line belonged to three communities in the part of building D [9]. It was fenced off from the longer, eastern part of the green space. The space was about 25 by 25 m. From the side of the building there was a green belt, in the middle there was a path around the lawn. In the corner of the building, there was a separate area with a carpet hanger, fenced with a hedge [53–55]. A row of probably three maple trees was planted at the southern border.

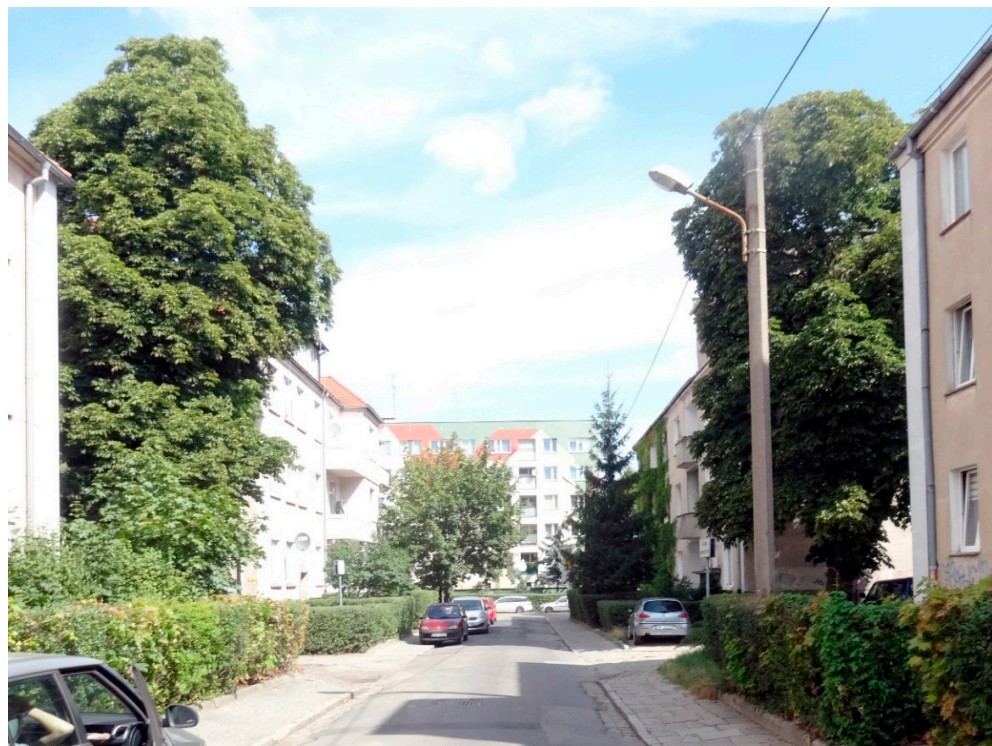

**Figure 11.** View along Stalowowolska Street towards the north. Chestnut trees closing the courtyard interiors. Summer 2019.

The last green space in this part of the estate is an elongated yard that is 115 m long and 40 m wide, located at building D, along Mielecka Street (Figure 12). From the side of the buildings, an escarpment ran along the entire length, moved away from the buildings by about 6 m [9]. It probably resulted from the necessity to level the area for construction. Thus, most of the green area was elevated in relation to the space next to the buildings by about 60 cm. It was possible to get to the higher placed area via stairs. In the northern part, the outline of the slope followed the arched outline of the facade, so that a small, irregular fragment with bushes was marked out from the rectangular interior [55]. The functional layout was realised in bands along the north–south axis: from the side of the buildings there was a recreational part, beyond a utility part with a lawn and places for drying linen and a zone of vegetable gardens adjacent to the back of gardens belonging to houses located on Tarnobrzeska Street [54].

The recreational part had a path running along the top of the slope and another one separating the site from the utility part. Along the path, three squares open to the west, with chestnut trees on the other sides were arranged. Two squares were 6 × 15 m in size; the last one was about 6 × 6 m. Six chestnut trees were planted on the larger squares, three were planted on the smaller square at its corners, perhaps emphasising the possibility of developing the composition towards the south. In addition, to access from the north–south path, the larger squares were accessed from the buildings through paths perpendicular to the facade.

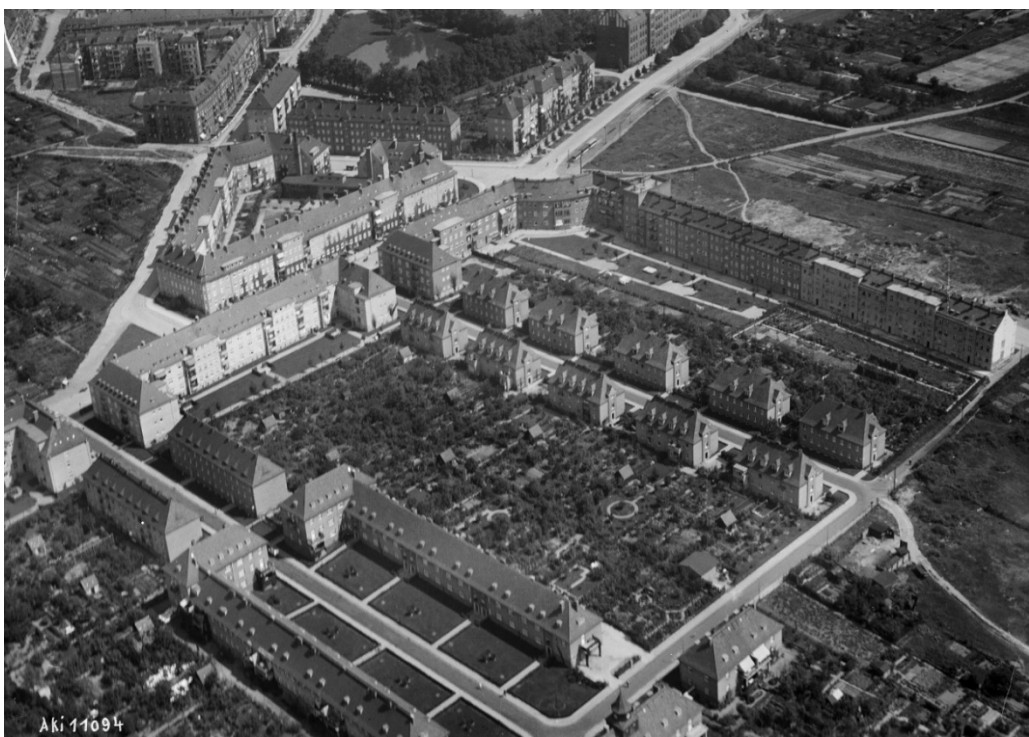

**Figure 12.** The aerial view at 'Am Sauerbrunn' estate towards the north-east in 1932. The courtyard at the intersection of Krucza and Mielecka Streets with three small squares encompassed by trees [55].

As in the case of the quarter at Kwaśna Street, in the cases discussed above, only the trees are a remnant of the pre-war development. The changes, the effects of which remain to this day, and which are visible in aerial photographs from the 1970s and 1980s [61,63,64], were of a degrading nature. Although it was necessary to adapt the courtyards to modern functions, e.g., waste storage and parking, it could be done in a thoughtful way. A glaring example is the last discussed yard, with a waste disposal site arranged in the middle, at the height of the middle pre-war square with chestnuts. Due to such a location, it was necessary to change the entire path system. The road was improved between chestnut trees, completely erasing the historical layout. In other cases, new functions were implemented—parking lots were introduced in the courtyards, next to playgrounds and waste disposal sites. As shown in Figure 9, the historical arrangement of the trees is incomplete. This is especially noticeable in the lanes of maple trees that grow in a narrow green strip where parking is carried out (Figure 13).

The third discussed area is the site at Wróbla Street, in the section between Generała Józefa Hallera Avenue and Sztabowa Street: both the space in front of buildings (A.3 in Figure 1) and the backyards (E.3a and E.3b). Its pre-war development can be studied and documented using surviving maps and drawings from the Wrocław Construction Archive [10,42], as well as an aerial photograph from 1932 from the collections of the Herder Institute [52] in comparison with the post-war photograph from 1947 from the Military Historical Bureau in Warsaw [60]. The buildings in this section of the street have been shaped away from the street. Both the houses located on Generała Józefa Hallera Avenue are set back from the road by about 20 m, and the street interior of Wróbla Street has been shaped in such a way that, while at the entrances, the walls of the opposite three pairs of buildings are 25 m apart, the middle nine pairs of buildings are separated from each other by 50 m. This allowed for the arrangement of wide green areas separated only by pedestrian circulation. The cadastral map from 1941 [10] shows that three of five of the lawns at Generała Józefa Hallera Avenue were shaped as terrain depressions, which is also visible in the post-war aerial photograph. The map also shows a double row of trees on

both sides of Wróbla Street, which is not confirmed by the photograph taken after the war. They were also not recorded on the land development drawing.

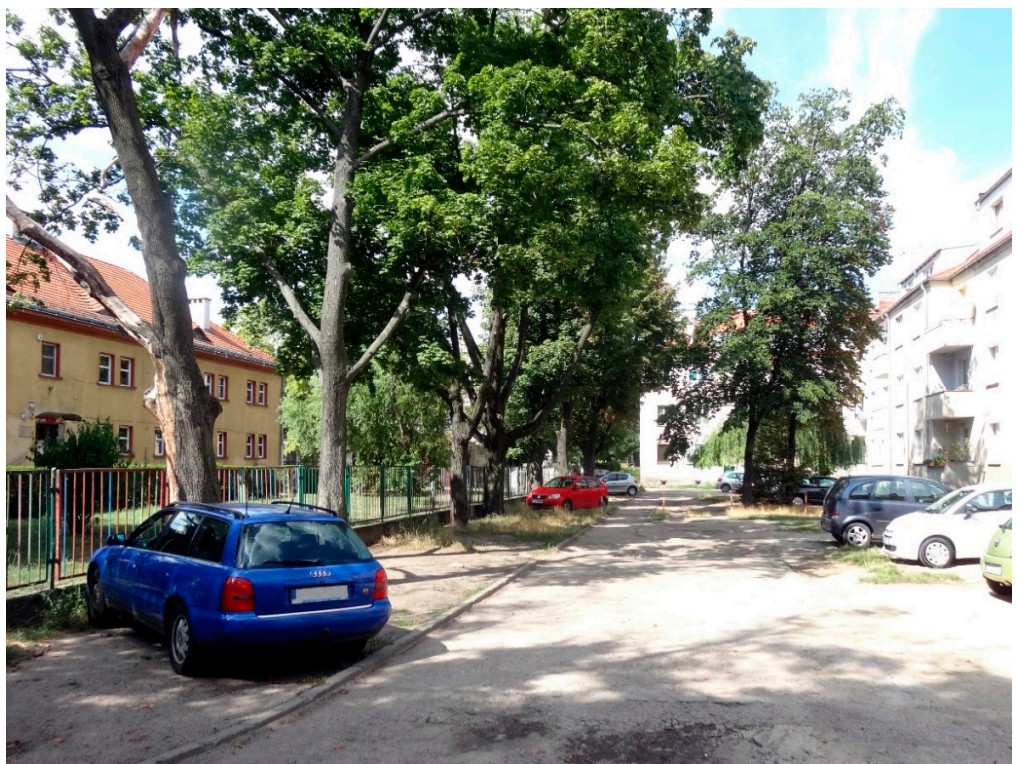

**Figure 13.** Backyard of building B designed by Heinrich Rump with a historical row of trees in summer 2019.

The development of the street space and courtyards is characterised by simplicity and functionality (Figure 4). Perhaps it is also a reflection of the crisis period at the turn of the 1920s and 1930s. Due to the fact that the building line was moved away, the yards belonging to the building were narrowed down to approximately 22 and 13 m or E.3a and E.3b, respectively. As in the case of the last described green area in the Am Sauerbrunn estate, also in this case, the terrain difference had to be solved with the help of a slope next to the buildings, which further reduced usable space. The slopes were planted with low shrubs, and the entrances were equipped with stairs [52]. Along the entire building, in the strip behind the escarpment, there were racks for laundry and carpet hangers. This part was edged with hedges. Wider spaces at the ends of the plot were also separated with hedges and arranged as neighbourhood spaces with sandboxes and benches. As the yard at the western side was wider, it was possible to distance the utilitarian part from the facade with the help of lawns. The paths heading to the back of the yard were accented with pairs of trees. They can be seen on the aerial photograph from 1947, as well as overgrown and not maintained hornbeam hedges, but it was not possible to determine the species, as these trees are no longer exist.

After the war, the wide spaces between the buildings at Wróbla Street were split in half to place parking spaces next to the buildings. Plantings along the street were implemented or restored, with ash trees on both sides, which had a positive effect on the aesthetic and ecological values of the street area. In the post-war period, the utility function of the courtyards changed—the laundry racks disappeared and waste collection areas were introduced instead. Interestingly, the recreational function of yard endings has been maintained for a long time. In the 1970s and 1980s, a parking lot was arranged only in the south-west corner [61,63]. The recreational areas have not survived [66]. Unmaintained

hornbeam hedges have grown, thanks to which the spaces between the buildings today have a unique character (Figure 14).

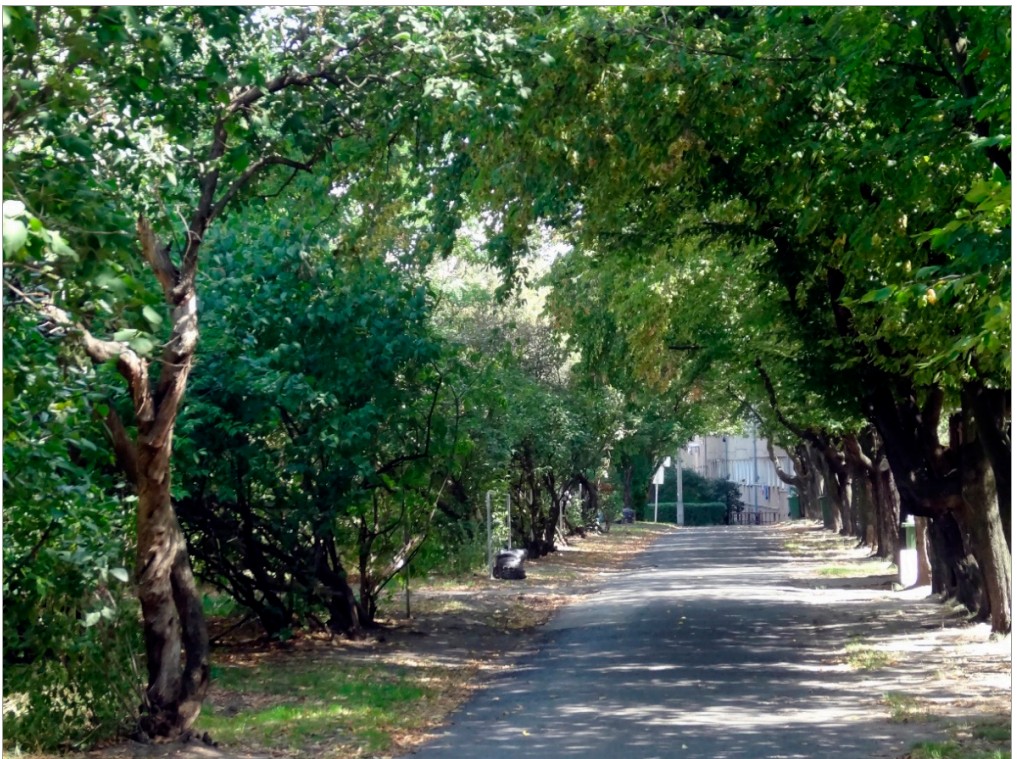

**Figure 14.** Backyard of the building by Wróbla Street, western part with overgrown hornbeam hedge. Summer 2019.

The final cases analysed here are two courtyards placed at the beginning of Wróbla Street (A.4 and A.5 in Figure 1). The land development was created in 1937 and 1938, respectively, and, in the author's opinion, did not meet the criterion adopted in the method of 'showing the features of a broader urban concept'. Nevertheless, they contribute to understanding the changes that occurred in the interwar period land development of the Gajowice estate, and thus are an important element in constructing the landscape biography of the estate. In both cases, it is only possible to compare the state as designed [43,44] with the current one [66]. Both of them show the features of an intimate ornamental garden development with utilitarian sections. The land development at the eastern side of the street was a courtyard with five houses. The site could only be entered through the buildings. The plot had a triangular shape, which affected the circulation layout: next to the buildings there were paths made of granite blocks, along the fence, the path was made of decorative broken stone. Near the buildings, in the middle of the path, there are places with carpet hangers, and at their ends there are sandboxes and a place to sit. In the middle of the establishment, there was a round square planted with four lime trees and three benches below them. The fence was planted with clumps of shrubs, in the southern part there was one additional tree, and the corner of the building was emphasised with a composition of shrubs (Figure 15).

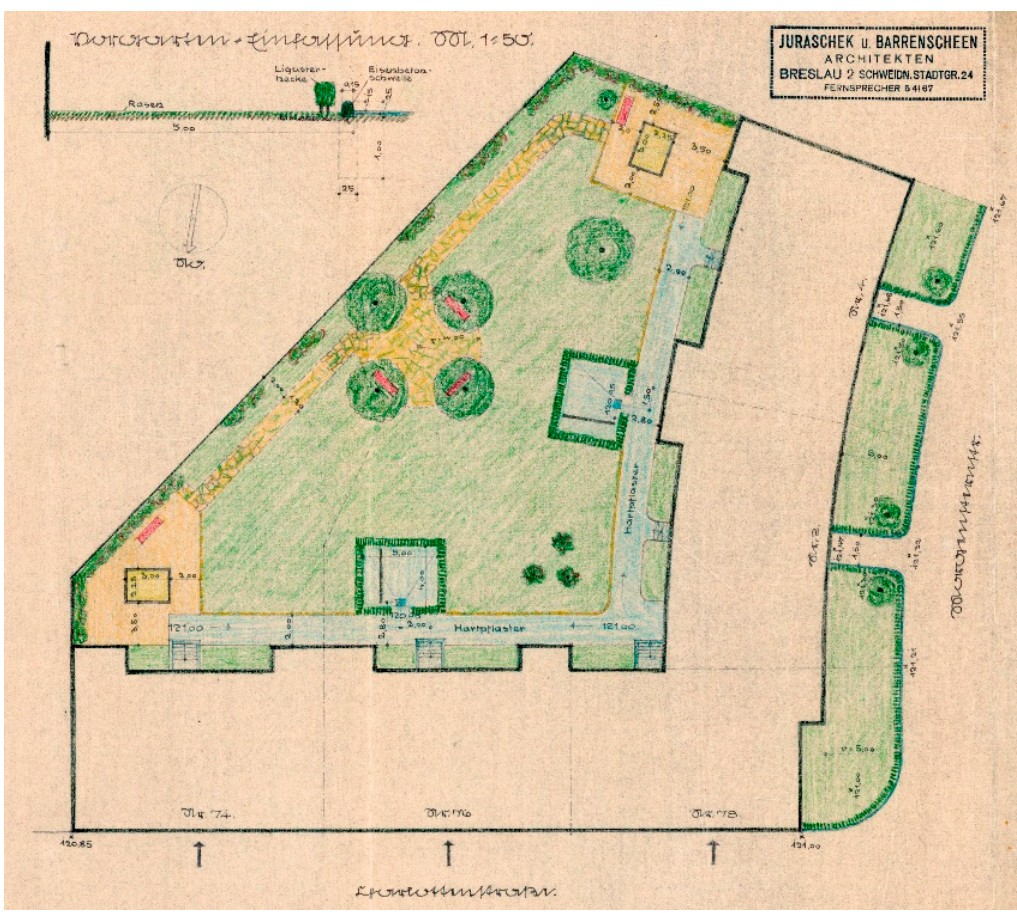

**Figure 15.** The design of the courtyard of the buildings at 2 Wróbla Street from 1937 by Juraschek u. Barrenscheen Architekten. Drawing south oriented [43].

On the opposite side of the street, the courtyard belonged to three houses. The plot was outlined in an irregular rectangle. In the corner, away from the buildings, an elevated part of a yard was made, emphasised by the corner planting of a bush or a tree. Below, behind a retaining wall, there are three benches on a path that surrounds the central lawn. The entire path was made of broken stone. The yard was fenced off with a formed hedge, and the entrance was only possible from buildings. At the facade of the building on the eastern side, there was a circular section with a diameter of 3 m with a bypass that was 2 m wide, planted with single shrubs, perhaps a sandbox. On both sides of Wróbla Street, the front gardens were 5 m wide. The entrances were accentuated with pairs of bushes, and the whole area was framed with privet hedges, as the description of the project shows.

The examples discussed here may indicate a change in the scale of urban planning assumptions, which reflects transformations of the interwar period economic situation. Wide ranging activity of construction companies was then past. Moreover, in this part of Wróbla Street, the whole urban concept was not completed before the war and was added to in the 1940s, but never fully completed. Regarding the examples of development under discussion, a composition of four lime trees encircling a round square is present to this day [66].

### 3.4. Current Neighbourhood Profile

The spatial development plan on a city scale, that is in force, defines the Gajowice as a part of a larger unit: Śródmieście Południowe [30]. The unit area is 378 ha, while the spatial scope of the study encompasses 53 ha. The land use data shows that residential areas predominate the unit: they occupy more than 30% of the total area. In 2016, the number of inhabitants was 52,700 people. Greenery sites occupy 33 ha, i.e., more than 8%

of the total area. The document states that the greenery system is rich and in different forms of arranged squares, semi-private greenery accompanying residential buildings, and street trees. Nevertheless, as judged by the authors, the unit faces many challenges concerning not only raising the standard of greenery in the areas of residence, but also creating systemic connections with green public sites. Challenges related to sustainable mobility are the development of rail transport and the creation of pedestrian and bicycle connections with neighbouring units.

Taking into account the needs expressed by the spatial development plan, in 2019, the city authorities established guidelines for the so-called complete estates, organizing participatory meetings, and working, among other things, on the case of the Śródmieście Południowe unit [31]. The aim of the consultation was to obtain feedback from the residents in four areas of interest:

- history and identity of the unit;
- character of public spaces and greenery;
- structure of local services;
- type of mobility within and outside the unit.

It should be noted that only 20 inhabitants of Śródmieście Południowe unit took part in the survey, which does not constitute a representative sample. Nevertheless, the response of the residents was analysed when determining the spatial dispositions for the complete estate.

The existence of local marketplaces and broadly understood post-German heritage were noted by responders in terms of the unit's identity. Regarding the second aspect, residents indicated that they are satisfied with the greenery in their surroundings, but noted numerous problems, such as damage due to illegal parking practices. They pointed to the lack of large parks or squares in the vicinity. The inhabitants were not able to clearly indicate the place of local integration and called for extending cultural, recreational, and gastronomic offers. During the consultations, the residents emphasized that their needs are met in terms of services. Regarding mobility, the discussed problem was the insufficient number of parking spaces and poor technical condition of sidewalks. At the same time, residents expected further development of the road network, including bicycle routes and public transport.

Another insight into the needs of the inhabitants gives the review of the participatory budget that has been established several years ago in Wrocław. This process enables residents to directly influence decisions to allocate part of the public budget to projects submitted annually by citizens. The review of the projects submitted in the last three years in the Gajowice [32–34] allows one to see the distribution of needs into different categories, which is summarised in Table 2.

**Table 2.** Participatory budget projects of Gajowice estate in the years 2019–2021, in the author's division into categories.

| Year [1] | Project Categorisation | | | |
|---|---|---|---|---|
| | Restoration of Courtyards, Incl. Green Infrastructure | Sports and Recreation Facilities in Public Areas | Improvement of Road Safety | Restoration of Grey Infrastructure, Incl. Bikeways |
| 2021 [2] | 3 | 2 | 2 | 1 |
| 2020 | 1 | 2 | 1 | 0 |
| 2019 | 4 | 2 | 1 | 2 |

[1] A different unit division before 2019 does not allow for data comparison. [2] Projects before evaluation by the municipality units.

As seen above, in 2019 and 2021, restoration of areas neighbouring the residential buildings represented the main category of submitted projects. The restoration of greenery within those projects is articulated, but the accents are distributed differently among elements such as car parks, playgrounds, blue infrastructure, green infrastructure, and other components of the spatial arrangement of the courtyards. It may indicate a wide range of needs and different issues experienced by the residents. The courtyard renovation projects

are followed by those that concern the issue of equipping public spaces, especially schools, with sports and recreational facilities. Projects of this type prevailed in 2020. Another important issue is the improvement of road safety around pedestrian crossings. The last category concerns ameliorating the quality of grey infrastructure, including cycling infrastructure. In fact, in 2021, one project concerned equipping bus stops with timetable displays, and in 2019, one concerned a bikeway and another car park in one of the courtyards.

## 4. Discussion

Urban concepts of healthy cities, taking into account the comfort of living and access to public spaces, derived from the Garden City Movement and written down in the form of planning regulations at that time, were verified by the experience of war, called by people then the Great War. The post-war period was characterised by large-scale development of urban planning in German cities, including Wrocław. The role of greenery as a factor shaping the space of human life was emphasised in the regulations [19,22,25], which was also reflected in architectural concepts, as these subjected to the study. In those assumptions, first attempts to treat greenery as a system that offers a number of benefits can be seen. The specialists of that time perceived green areas as an element shaping social equality and individual's freedom, postulating access to one's own piece of land for all [24]. However, in the case of areas intended for temporary gardens in the Gajowice, perhaps one should speak of necessity rather than economic self-sufficiency. The crisis of the late 1920s slowed down the pace of extensive changes; however, the idea of a green and accessible city, but also an affordable flat is still valid today.

The development of the southern suburbs of Wrocław, which started in the middle of the nineteenth century, is a history of the transformation of agricultural land into an urban area. Nevertheless, the spatial planners at the turn of the century paid attention to not only the functional layout of the unit, but also the place's context. An urban continuum is visible in practices such as respecting the previous road system or transforming a place with its own identity into the district's main recreation site, as on the example of Sour Source. In the spatial arrangement practices noted in the Gajowice estate before the Second World War, features of perspective planning could be seen. The streets had been designed with an appropriate width, so that after the construction of the buildings, trees were planted. An adequate reserve was left for the tram line extension. Various forms of urban mobility such as public transport and bicycles were taken into account. The surviving documents contain transformation's plans of a private plot into publicly accessible green areas. On the other hand, it shall be noted that some areas were built-up, although originally there, had been intended for greenery or temporarily for allotment gardens or recreational sites. Nevertheless, the Gajowice greenery system of the interwar period encompassed different forms of green infrastructure: street trees and hedges, urban park and playgrounds, allotment gardens, and courtyards rich in various forms of vegetation.

Research on the original land development showed that the designers of the interwar period adopted principles based on not only compositional or functional aspects, but also recognized the natural values in the variety of green forms used in one place, including wall climbing plants, and the important role of trees with large target sizes. It can be seen that the greenery system was planned as to permeates urban tissue. Looking from a contemporary perspective, those solutions could be identified as nature-based, designed to create ecological corridors and positively influence the local climate. Another important observation is that trees are a stable element of the cityscape; they are often only remnants of a historical layout. An overview of the land development adjacent to the buildings from the 1920s to 1930s in the Gajowice estate shows the transition of courtyards' functional and spatial programs: from complex ones to more simple utility forms and even small ornamental gardens. That probably reflects the overall economic situation of a country, the context that has already been recognized in other publications [25,35]. Nevertheless, in almost all studied cases, the recreational function of the courtyards played a major role.

The study shows that post-war changes resulted in the blurring of the green system and designers' original assumptions, as evidenced by the numerous transformations of large green areas of the Gajowice estate, such as the park at Sour Source, a school garden, or the former gardening school. It should be added, however, that sometimes the lack of maintenance—as in the case of numerous hornbeam hedges—influenced the development of large trees, that have higher value than shrubs when viewed through the prism of natural assets. Mostly, however, the depletion of green sites is visible: only some of the trees are left of various greenery forms in the courtyards. The courtyards are mainly used for utility needs related to waste storage and parking, which was not the original intention, neither the design nor the legal one. On the other hand, it partly reflects contemporary requirements.

The current needs of the unit's inhabitants, included in the spatial policies and expressed in the participation processes, although broader than only related to the quality of greenery, are referred to in many aspects. The issue is not only the poor standard of green sites in the areas of residence, but also the scarcity of systemic connections among them. Moreover, the inhabitants are not able to clearly indicate the place of local integration. They notice the lack of large parks or squares in the vicinity. Interestingly, post-German heritage is perceived by the residents in terms of the unit's identity. Nevertheless, it seems that the variety of needs requires a rational approach to the planning of 'green' policies.

## 5. Conclusions

This study of a landscape change of the Gajowice estate showed that many elements of the greenery system were planned with reflection and the long perspective of sustaining the dwellers areas of recreation. Various forms of greenery were used as coverings for building walls, partitions, and fences, and finally as elements determining the local climate, shading streets, and squares. Therefore, they should be seen, in addition to the buildings, as an essential element of cultural heritage, and in combination with their environmental properties, they shall be treated as green infrastructure. In the face of the effects of the climatic crisis, it is evident that it is not enough to develop based on the relationship between man and inanimate nature, but it is necessary to also take biological diversity into account. Looking from the perspective of adverse changes, the conservation of the historical system's remains seem more justified. Steps need to be taken to preserve its remnants and restore the system's continuity. The land development character shall be adapted to the contemporary requirements of cities that are resistant to climate change. Spatial solutions examined with the help of comparative cartographic studies could become an introduction to a catalogue of local solutions, supporting biological diversity and human health, that is based on cultural heritage.

**Funding:** This research received no external funding.

**Data Availability Statement:** Not applicable.

**Acknowledgments:** The author would particularly like to thank the Wrocław Construction Archive of the Museum of Architecture and Herder Institute for Historical Research on East Central Europe for consenting to the use of iconographic materials from their collections that helped to illustrate and greatly enriched the article.

**Conflicts of Interest:** The author declares no conflict of interest.

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
