# Peer review of "Towards an Understanding of the Pre-War Landscape Transformations in the Face of Contemporary Urban Challenges on the Example of Gajowice in Wrocław"

_sustainability, doi:10.3390/su13115962_

Round 1
Reviewer 1 Report
The article focuses on the construction and transformation of the neighbourhood of Gajowice designed at the beginning of the last century in a rural area west of Wroclav, part of the Weimar republic at the time, and implemented after WW2.
With reference to few cases of the neighborhood, the work deals with the level of permanence or change in open space and in ‘greenery’ compared to the original project, contending that, differently from previous studies focused on the built environment, its inherent originality stays in its depth insight over the green areas seen as a ‘ante litteram green infrastructure’.
The main remarks and suggestions are the following:
'Sustainability' magazine emphasizes connections in theory and practice among sustainability discourses and issues.
These standpoints are completely missing.
The 'Sustainability' reader's interest in the topic presented by the paper could be raised by an introductory review of the main features characterizing coeval planning experiences such as Berlin and other German cities that laid the foundations of modern urban planning.
The work should emphasize the pioneering insights of the planners and architects.
Also missing is current neighborhood profile: who dwell in Gajovice, how many inhabitants, are they happy with the provision of public space, and their everyday life. Accordingly, a reading only of the 'greenery' layer does not allow for current needs. And how the current management works, at the local scale.
The analytical descriptions relating to the survival of tree species case by case do not add to the work.
Finally, since there are no limits to pictures, the study would take advantage of a more complete iconographic apparatus.
Reviewer 2 Report
Dear Authors
The manuscript is interesting, but the structure needs adjustments. The abstract is incomplete with research methodology, findings research and conclusion lost.
The introduction section is also incomplete and it is necessary to include the question-problem and the objective of the study, as well as to clarify the problematization and justifications of the research. Everything is mixed in the way it was submitted and this causes difficulties to understand the problems and justifications of the research.
The method section can be more detailed so that other researchers understand how the study was done.
Reviewer 3 Report
The interesting topic of the work fits well with the subject of the journal. Interesting work, full of detailed descriptions. I became acquainted with it with pleasure. However, I have some reservations of a formal nature and related to the structure of the work, which are discussed below.
- Abstract does not meet the journal's requirements. It is 25% too long in relation to the maximum length indicated in the recommendations for Authors. The subject of research is indicated, but in the background I did not find the clearly indicated purpose of the work. The methods are only shown at the end of the abstract. Please read the guidelines for authors and organize your abstract.
- "Urban adaptation" was written in "key words", this term was used only once in the whole work (in key words); landscape development (2x); historical green infrastructure (3x); urban landscape (1x); landscape planning (2x); landscape biography (3x). It seems to me that key words should correspond to the content of the work, which is not the case here. Does not understand how key words can characterize work, since they are not used in it.
- In in the "Introduction" part, I did not find a clearly formulated aim of the work or research hypothesis.
- The work was not set in the context of similar research, which may indicate its innovative approach. However, to some extent this is contradicted by works that can be found, for example, using the links below. It may be helpful in creating a context for the conducted research, which, in my opinion, is missing from the work being assessed
https://doi.org/10.1016/j.landusepol.2019.104236
https://bazhum.muzhp.pl/media/files/Bulletin_of_Geography_Socio_Economic_Series/Bulletin_of_Geography_Socio_Economic_Series-r2011-t-n15/Bulletin_of_Geography_Socio_Economic_Series-r2011-t-n15-s117-129/Bulletin_of_Geography_Socio_Economic_Series-r2011-t-n15-s117-129.pdf
- “Discussion” - in relation to the detailed elaboration of the "Result" part, it leaves a significant lack of satisfaction. In fact, in my opinion, there is no scientific discussion in this part of the paper.
- The “Discussion” and “Conclusion” combination is eligible - see information for authors (see point 5. Conclusions. This section is not mandatory but can be added to the manuscript if the discussion is unusually long or complex), but in this case, in my opinion, we have no problem “Unusually long or complex” discussion. I believe that for the quality of all work, discussions should be written in accordance with the journal's guidelines, and the conclusions should be placed as a separate part. In the current version, these parts are mixed, and some statements, for example in line 646, not supported by earlier analysis and not included in the text. Many aspects presented in the "Result" section were not discussed in the discussion, and some were not discussed in relation to the literature on the subject.
- Relatively modest literature on the subject. Out of 52 items listed in the reference list, 38 are plans and photos. It proves well the material collected by the author. On the other hand, however, it shows the weakness of the theoretical approach and literature on the subject in general as well as in detail. In my opinion, the subject literature should be expanded.
- Some of the presented materials (photos, scans of plans) of poor quality.
- Detailed comments:
Line 139-141. There is "After the First World War, Wrocław was the most populous city of the Weimar Republic. The population achieved 114 people per hectare of the total urban area, compared to 46 major cities, with the average number of 41.3 people per hectare. [14]. " The quoted work refers to Silesia, not to the Weimar Republic. Sorry, but I don't understand these data and calculations. According to the data available on the internet:
The population of Berlin in 1919 (December): 1,928,432; the area of the city in the same year: 66.93 km2, the population density: 28,813 inhabitants per 1 km2.
The population of Wrocław in 1919 (December) 528,260; the area of the city in 1924 (it hardly changed compared to 1911): 49.61 km2; is the population density: 10,648 inhabitants per 1 km2.
Line 638. There is "It could be said that the post-war exchange of residents broke the continuity of understanding of the city's green system,". This thread has not been emphasized in the conducted research, and should not be included in the discussion / conclusions without prior analysis.
Line 646. There is "A study of a landscape change of Gajowice estate showed that many elements of the greenery system are not in good condition,…". This conclusion, which is probably legitimate, does not result from the "Result" section.
Round 2
Reviewer 1 Report
The paper has been substantially improved.
A further effort by the author should be to summarize in what the original layout of the neighborhood can be seen as a 'proxy' of an eco-district. Indeed, 'perception' is a vague term that requires cautiousness.
Author Response
The Reviewer’s comment has been carefully read by the author and resulted in additions to the text. Please see the attachment.

Reviewer 3 Report
I appreciate the effort made to improve the article, shortened abstract, keyword improvement and other changes. Especially in terms of formulating and presenting the aim of the work and the research hypothesis. Also extensions to the sections "Materials, Methods", "Discussion" and "Conclusion".
However, I still have a few comments about which below.
1. There is still no research purpose in the abstract.
2. In key words: we have "historical green infrastructure" that appears in the text, in the conclusions only once, just like "Wrocław's urban development". "Landscape changes" was also used only once in the text; and "landscape biography" only twice.
3. I am not entirely convinced that the purpose of the research "The aim of the paper is to contribute to knowledge of the original assumptions of the development of areas adjacent to multifamily buildings, bearing in mind the wider urban planning context of the interwar period" justifies their undertaking .
4. If I understand it correctly, the items given in the literature [17,18] present the knowledge about the original assumptions for the development, including the area under study, in the broad urban context of the interwar period. Which may lead to the conclusion that the prepared work is redundant from the point of view of achieving the goal.
5. Regarding the research hypothesis: "The research hypothesis states that due to the fact that these estates were planned with great care for the buildings' surroundings, the used solutions would now be defined as green infrastructure or nature-based solutions" in my opinion also does not require undertake research. A good landscape architect, after visiting the studied part of the city, when asked about the adopted solutions for the distribution of greenery (trees), would probably (define) them as meeting the definition of "green infrastructure or nature-based solution", i.e. similar to the author of the article.
6. The conclusion of the sentence with the hypothesis "which was confirmed by the study" should probably reach the conclusions at the end of the work with appropriate justification resulting from the analysis of the material using the given research methods and after the discussion of the issue.
7. It also seems to me that we are dealing with a lack of consistency between the stated purpose of work and the title of the job, or its not fully precise definition. In the title, we have understanding in the contemporary context (line 2-4) "Towards an understanding of the pre-war landscape transformations in the face of contemporary urban challenges on the example of Gajowice in Wrocław". ) "The aim of the paper is to contribute to knowledge of the original assumptions of the development of areas adjacent to multifamily buildings, bearing in mind the wider urban planning context of the interwar period".
8. While appreciating the development of part 3, I have doubts related to part 3.4 Current neighborhood profile. He does not understand how the 2019 residents research is related to the planning of the district from the interwar period. Their needs, indicated on the basis of the participatory budget for 2019-2021 (Tab. 2, line 703), are in my opinion ahistorical in relation to the needs of the inhabitants from the period of the development of the studied area.
9. The conclusions devote almost the entire paragraph to the request for "cultural heritage" and "historical green infrastructure" (line 776-788), which does not result from the analysis of the collected materials and discussions in which cultural heritage appears twice (and only in the conclusions). green infrastructure not even once. In my understanding, such conclusions do not result directly from the conducted research.
10. Minor technical shortcomings, for example:
Line 82: No space before title "Materials ..."
Line 808: year-bold, also no bold selection in the years of publication in the next dozen or so works in the literature list
Line 810: year - bold, which document exactly?
Line 813: these are two pages of an information leaflet.
Author Response

(The authors gave the same response as above.)

Round 3
Reviewer 3 Report
The improvements made in terms of refining the goal and hypotheses make the text better in my opinion. The doubts presented earlier were clarified. Thank you. However, I have a few more comments which I understand will improve the text.
1. In the Discussion part, despite its improvement, which I appreciate, there is still no discussion / confrontation with the results of research on this topic by other authors. In this way, you can show the achievement of new results. Unless this is the first work to capture the topic in this way, as there are no similar works in Discussion. The author seems to be arguing with himself.
2. In Conclusion, part of first paragraph "The rules of thumb implemented by urban planners of the interwar period are now raised as a necessity by initiatives such as the European Green Deal and its derivatives, such as the New European Bauhaus;" (line 794-796), in my opinion it does not result from the research You have undertaken. The New European Bauhaus appears as an idea in the Introduction, we can learn about it from the advertising leaflet, and then it appears in the Conclusions. It is similar with the European Green Deal. I am not saying that it is not so. I only claim that even a real belief (opinion) in a research work requires confirmation with evidence, analysis and discussion from which Conclusion results. I appreciate the expansion of the text and adding the ref. [8], but I propose to clarify it as [8] (pp. 65-69). However, there is nothing in this work about the New Green Bauhaus. But maybe he misunderstands this piece of work. Of course, all considerations about green areas are in some way related to the European Green Deal and possibly the New European Bauhaus, but in my understanding of the text it does not result from the work.
3. Despite the author's explanations regarding ref. [7] on the New European Bauhaus, I insist on replacing it with another - or considering removing it (along with an entire paragraph of the work on the subject, also from Abstract and Conclusion) (see ref. 2).
Author Response
After getting acquainted with the valuable critique, author has implemented the changes. Please see the attachment.
